# MobileAgentBench: An Efficient and User-Friendly Benchmark for Mobile LLM Agents

## Abstract

Large language model (LLM)-based mobile agents are increasingly popular due to their capability to interact directly with mobile phone Graphic User Interfaces (GUIs) and their potential to autonomously manage daily tasks. Despite their promising prospects in both academic and industrial sectors, little research has focused on benchmarking the performance of existing mobile agents, due to the inexhaustible states of apps and the vague definition of feasible action sequences. To address this challenge, we propose an efficient and user-friendly benchmark, **MobileAgentBench**, designed to alleviate the burden of extensive manual testing. We initially define 100 tasks across 10 open-source apps, categorized by multiple levels of difficulty. Subsequently, we evaluate several existing mobile agents, including AppAgent and MobileAgent, to thoroughly and systematically compare their performance. All materials will be accessible on our project webpage, contributing to the advancement of both academic and industrial fields.

## 1 Introduction

With the emergence of large language models (LLMs) (Achiam et al., 2023), researchers have developed various autonomous agents across fields such as robotics (Bousmalis et al., 2023; Reed et al., 2022), games (Wang et al., 2023b; Du et al., 2023), and mobile phones (Yang et al., 2023b; Rawles et al., 2023). Among these, mobile agents have attracted significant attention due to their potential to enhance user experiences and provide intelligent assistance on-the-go.

People have been dreaming of Intelligent Personal Assistants (IPAs) (de Barcelos Silva et al., 2020) that can fully automate daily tasks for decades. Since Apple introduced its digital assistant, Siri (Apple, 2011) in 2011, almost all the leading technology companies have launched their own IPAs, including Microsoft Cortana (Microsoft, 2014), Amazon Alexa (Amazon, 2014), and Google Assistant (Google, 2016). While these digital assistants provide a hands-free human-computer interaction experience using Natural Language Interface (NLI), they can only fulfill relatively simple tasks, such as setting an alarm clock or sending a text message (Li et al., 2024). For third-party apps, developers have to follow and implement the application programming interfaces and protocols, such that when a user issues a very specific command, the system can invoke the corresponding functionality. This limits the usability of those digital assistants.

LLMs contribute significantly to resolving the persistent challenge of understanding user intent. The demonstrated reasoning ability (Qiao et al., 2022) highlights the potential of LLM-based autonomous agents as next-generation Intelligent Personal Assistants (IPAs), which are not limited by the programming interfaces since they directly operate on the Graphic User Interface (GUI) (Wang et al., 2024; Yang et al., 2023b). The GUI can either be represented by a text-based view tree to be consumed by a LLM or a screenshot image that can leverage a Multi-modal LLM (MLLM) (Yin et al., 2023). The action space of the agents composes a series of functions to simulate human operations, such as click, type, swipe, etc. In this way, LLM agents can theoretically achieve whatever human users can do, without any modification of the existing apps.

The promising future of LLM-based smartphone agents attracts more and more researchers to study this topic. However, the scope of benchmarks available for evaluating the performance of these agents remains constrained. Among the existing benchmarks, several prevalent issues are evident:
**1. Scalability and usability**. Researchers need to fully understand complicated data structures and

Table 1: Comparison between the proposed and existing benchmarks

| Benchmark | Fully Autonomous | Realistic Environment | Flexible Success Condition | Low Code Invasiveness | Large Scale |
|---|:---:|:---:|:---:|:---:|:---:|
| AppAgent (Yang et al., 2023b) | ✗ | ✓ | ✓ | ✓ | ✗ |
| AITW (Rawles et al., 2023) | ✓ | ✗ | ✗ | ✗ | ✓ |
| AndroidArena (Xing et al., 2024) | ✓ | ✓ | ✗ | ✗ | ✓ |
| AppBuddy (Shvo et al., 2021) | ✓ | ✓ | ✓ | ✗ | ✗ |
| AndroidEnv (Toyama et al., 2021) | ✓ | ✓ | ✓ | ✗ | ✓ |
| B-MoCA (Lee et al., 2024) | ✓ | ✓ | ✓ | ✗ | ✗ |
| AndroidWorld (Rawles et al., 2024) | ✓ | ✓ | ✓ | ✗ | ✓ |
| LLamaTouch (Zhang et al., 2024) | ✓ | ✓ | ✓ | ✗ | ✓ |
| MobileEnv (Zhang et al., 2023) | ✓ | ✓ | ✓ | ✗ | ✓ |
| MobileAgentBench (ours) | ✓ | ✓ | ✓ | ✓ | ✓ |

tools before extending the benchmark with customized tasks or integrating it into their own code-bases (Zhang et al., 2024). **2. Robustness and Flexibility.** Only the annotated task completion path is considered (Rawles et al., 2023; Xing et al., 2024). However, there might be multiple paths to successfully complete a task, which may break the task success judgment logic. **3. Realistic environment**. Some benchmarks evaluate the agent's performance based on a collection of screen-shots but not real devices. It fails if the agent performs an abnormal action and goes to an undefined state (Rawles et al., 2023).

To address the issues described, we propose a robust benchmark, MobileAgentBench, designed to evaluate the capabilities of mobile LLM agents within the Android ecosystem. MobileAgentBench offers several advantages over previous benchmarks due to its ease of use and minimally invasive nature. Specifically, for standard agents, the integration process requires fewer than ten lines of additional code. The benchmark excels in usability and versatility, supporting a broad spectrum of testing tasks across various Android operating system versions and executing on actual devices.

In this initial release, we offer 100 built-in benchmarking tasks spanning ten open-source applica-tions. Notably, MobileAgentBench diverges from traditional approaches by simplifying the exten-sion process. Third-party developers can specify the conditions for task success using just a few lines of Python code, without needing extensive knowledge in Android development. This accessi-bility makes MobileAgentBench more conducive to developers and researchers from non-Android Development communities. Furthermore, we introduce an innovative method for determining the task-terminating state, rendering the benchmark resistant to the complexities of tracking multiple potential success pathways. This approach ensures that MobileAgentBench provides reliable and precise benchmarking outcomes.

The comparison between the proposed and existing benchmarks is listed in Tab. 1, where fully autonomous represents if the benchmark does not need human supervision or judgment, realistic environment represents the tasks can be run on real devices, success condition flexibility represents it takes all possible success paths into consideration, low code invasiveness represents integrating the benchmark into existing agents does not need significant code changes.

Our **contributions** are summarized as follows:

- We propose a benchmark framework for mobile LLM agents. The new approach addresses com-mon issues of existing benchmarks, making the evaluation process fully autonomous and reliable.
- We implement and test 100 benchmarking tasks with different levels of difficulty. The benchmark is plug-and-play and easy for both developing new agents and evaluating existing agents.
- We evaluate the performance of state-of-the-art mobile LLM agents and perform a solid and sys-tematic comparison with our new benchmark, providing baseline data to the community.

## 2 RELATED WORK

### 2.1 MOBILE LLM AGENTS

Studies before the LLM-era employed reinforcement learning (RL) to solve the autonomous GUI navigation problem (Gur et al., 2018). The recent advancement of LLM and MLLM becomes the dominant agent paradigm becaust of the greater ability of UI understanding and reasoning. Early studies focused on web agents, which achieve task automation within browsers (Deng et al., 2024; Zhou et al., 2023). Recently, more and more studies started to investigate agents on mobile devices, especially on the Android platform, as Android smartphones are the most widely used personal computing devices.

Mobile LLM agents share a similar algorithm. The full input prompt often consists of four main components: the user prompt (task description), the current UI view hierarchy (VH) description, the action function list, and historical information. Specifically, the action list mainly includes click, swipe, type, and other common UI operations. If MLLM is used, the current screenshot is also a part of the input. The LLM/MLLM is asked to think of the next action based on the current and historical states and call the correct function to perform the given task step by step. The agent finally parses the model response and sends control signals to the Android device using Android Debug Bridge (ADB) [1], UIAutomator [2], or other higher-level UI automation frameworks.

Despite the similarity of the high-level ideas, researchers have developed different techniques to improve performance and efficiency. Among these works, AndroidArena (Xing et al., 2024) transforms the long and overwhelming view hierarchy XML into a compressed representation and assigned UI elements with unique node IDs, which shortens the prompt and makes the system more efficient. MobileAgent (Wang et al., 2024) observes that GPT-4V lacks the capability of UI element localization, and employs an Optical Character Recognition (OCR) model to locate and localize text views. Moreover, it uses the CLIP (Radford et al., 2021) and Grounding DINO (Liu et al., 2023) models to detect icons. AppAgent (Yang et al., 2023b) uses SoM (Yang et al., 2023a) prompts to localize UI elements and breaks tasks into two phases, exploration, and deployment. During the exploration phase, the agent automatically interacts with the apps and summarizes the observations into a document. In the deployment phase, it employes the Retrieval Augmented Generation (RAG) (Lewis et al., 2020) technique to utilize the summarized knowledge and improve success rate. CogAgent (Hong et al., 2023) proposes its own highly efficient 18B-parameter MLLM, which can be loaded on a single commercial GPU. Furthermore, Octopus v2 (Chen & Li, 2024) proposes a compact 3B-parameter model, which unlocks the potential to run mobile LLM agents on-device in an efficient and privacy-preserving manner.

### 2.2 BENCHMARKS FOR MOBILE LLM AGENTS

Since the Mobile LLM agent is a newly emerging research field, the choice of benchmarks is very limited. Some studies rely on verifying the task execution status manually to evaluate the performance (Yang et al., 2023b), which is tedious and time-consuming. To expedite the agent development, we need a fully autonomous benchmarking system to report various metrics, especially the task success rate. However, automatically judging if a task is completed successfully is non-trivial. The main challenge is caused by the dynamic nature of the GUI navigation task – the agent may perform random actions and drive the app to an unknown state.

AITW (Rawles et al., 2023) is a popular benchmark for mobile LLM agents. It has a large scale, but it's based on static screenshot images. Thinking of each app state (screenshot) as a node, and each action as an edge, we can build a State Transferring Graph (STG) based on the screenshots and the human-annotated actions. It fails immediately if the agent performs actions in a non-considered sequence and leads the STG to a non-existent node, even if the agent can eventually complete the task.

The only solution is to identify task successes on real devices. One approach is to match the agent's actions with the annotated ground truth (GT). A step-wise matching algorithm is not accurate, because the agent may not finish the task exactly in the same order with GT. AndroidArena (Xing

---

[1] https://developer.android.com/tools/adb
[2] https://developer.android.com/training/testing/other-components/ui-automator

et al., 2024) proposes an adaptive method of calculating the longest common subsequence of the agent and GT action sequences, which is illustrated as follows, where $a$ and $\hat{a}$ are the GT and actual actions, respectively. The common subsequences are marked in red.

$$a = ABC \tag{1}$$

$$\hat{a} = AXYBUCVW \tag{2}$$

AndroidArena (Xing et al., 2024) treats a task as successful if the GT is a subsequence of the actual action sequence. It addresses the issue of redundant actions but is still not optimal. A simple counter-example can be navigating from the page 1 to page 3 by clicking the next page button two times. If the agent performs the following sequence of actions: clicking the next page button, clicking the previous page button, and clicking the next page button, it doesn't navigate to the correct page but is still a subsequence of the GT. This method gives false positive results if the redundant action has a side effect.

A concurrent work, LlamaTouch (Zhang et al., 2024), addresses this problem by examining the final UI state, which is similar to our approach. We observe that despite the infinite feasible action sequences, the final success state convergence to one. The success or failure can be determined by checking the final UI state. An edge case is that some tasks may not have a direct UI representation, for example, the result of a network request triggered by a button may not be directly reflected on the current UI page. Thus, only checking the UI state is not sufficient and we need to incorporate actions, such as the clicking event, into consideration. LlamaTouch (Zhang et al., 2024) matches the click action by mapping the coordinate to a UI element, based on the view bounding boxes. However, it may not always be accurate. The process of finding the correct view to respond to a clicking event is called a hit test, and it's only accurate if performed by the Android UI system. This is because app developers can modify the touchable area, making it different from the view border to get better user experience.

Fig. 1 shows an example of enlarging the touchable area of a button view. In Fig. 1, touching point 1, the button does not respond because it's outside of the touchable area. Touching point 2, the button responds because it's inside the button view. Touching point 3, although it's outside of the visible button view, the button still responds because it's inside the extended touchable area. To overcome this difficulty, we utilize the Android Accessibility Service [3] to capture app events faithfully and forward them to the benchmark server. The details of our implementation are described in Sec. 3.1.

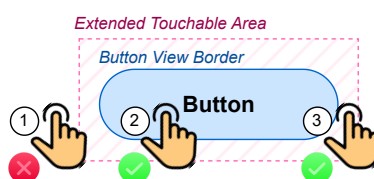

Figure 1: Extended touchable area.

Another concurrent work, AndroidWorld(Rawles et al., 2024), offers 126 tasks across 20 apps and has its own agent, M3A, which is benchmarked against others using automatic evaluations and precise rewards. We all emphasize the importance of real-device benchmarks to truly reflect performance and employ Python to gauge task success. This enhances flexibility and accuracy by capturing dynamic system states instead of relying on static UI matching. However, our approach differs significantly: we prioritize agent-centric development while AndroidWorld focuses on benchmark-centric strategies. Our MobileAgentBench integrates seamlessly with minimal code adjustments, supporting existing agents' action spaces through the Android Accessibility Framework. This enables less intrusive adaptations and broader evaluations beyond mere success rates, facilitating a more detailed assessment.

## 3 MOBILEAGENTBENCH

### 3.1 METHOD

MobileAgentBench runs on real Android devices, supporting both physical devices and emulators. It sets up environment and then invokes the agent execution function. While the agent is operating

---

[3] https://developer.android.com/reference/android/accessibilityservice/AccessibilityService

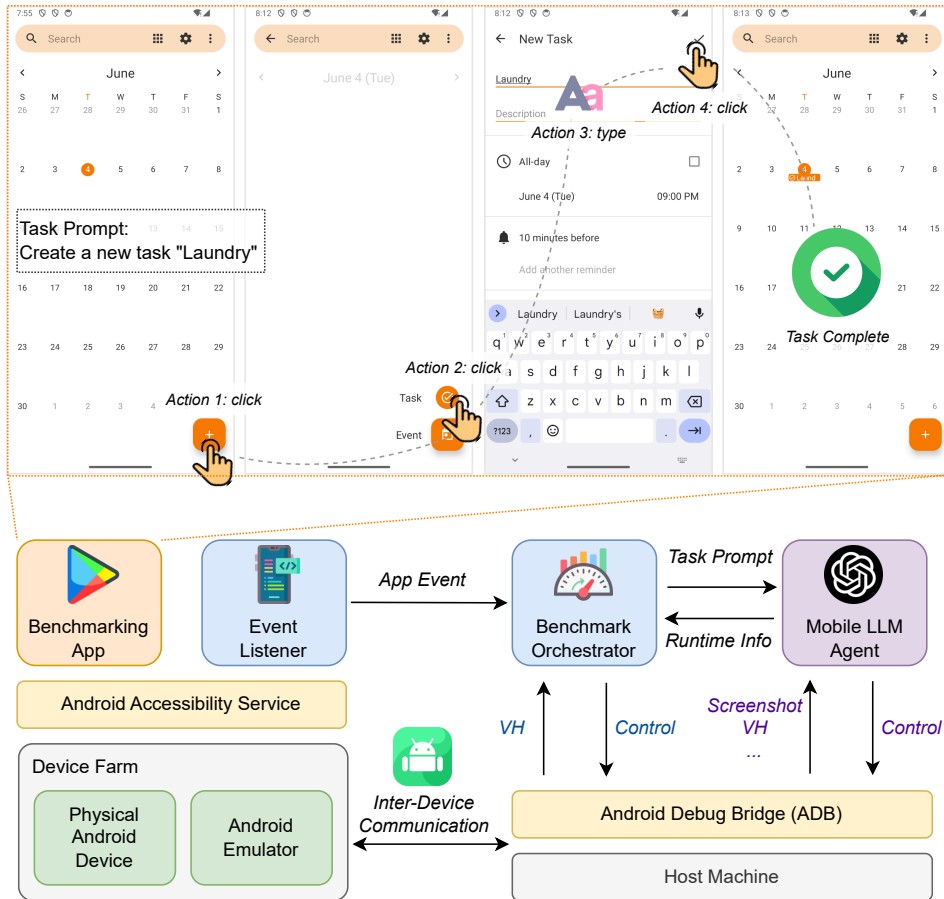

Figure 2: Overview of the MobileAgentBench architecture.

the device, MobileAgentBench judges the task success status in real time without any side effects. After the agent stops or exceeds the maximum steps, MobileAgentBench automatically switches to the next task. The whole process is fully automated and requires no human supervision.

The task success judgment mechanism is implemented by matching the final UI state, instead of examining the action sequence. This is because there might be multiple paths towards task completion, but the final success state converges to one. For example, if the task is to go to the settings page, agents may mistakenly open random pages before they correctly find the desired settings page. Matching the action sequence is difficult because of the randomness. On the contrary, checking if the top page is the settings page is easy and reliable. No matter what operations the agent does, we treat it as a success as long as the settings page is detected. ADB and UIAutomator are used to fetch the VH information. For each task, there is a `Python` file that defines the task success criteria, making it easy to extend and customize tasks for third-party developers.

As some tasks may not have a direct reflection on the current UI page, only checking the VH information is not sufficient. An example can be editing a note and saving the changes. Clicking the save button, the app may only pop up a temporary alert to indicate the save action has succeeded and stays on the current page. When the benchmark checks the current view state, it doesn't know if the save button is clicked or not. As a benchmark, it cannot go to other pages because it may change the app state and affect the next action of the agent. Since we want to determine the task success in real-time to collect how many steps the agent takes, it is not feasible to check UI states of other pages after the agent stops. Besides, some agents have the problem of not being able to stop gracefully even the task is completed. We address this issue by incorporating app events, especially button click action signals. For the above-mentioned task, we can use the view hierarchy to check if

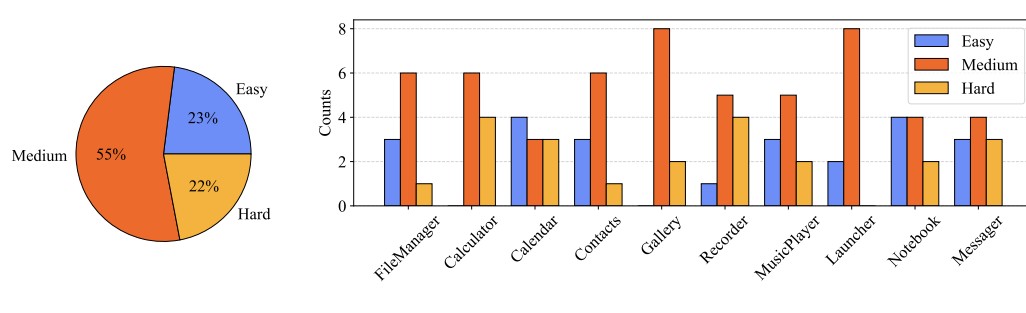

(a) Distribution over all tasks.                    (b) Distribution over each app.

Figure 3: The distribution of task difficulty levels.

the note is edited correctly, and then mark the task as a success if the save button clicking signal is received afterwards.

To faithfully receive the app event signals, we make an Android app using the Android Accessibility Service. Android Accessibility Service was originally designed to help people with disabilities. It runs in the background and invokes a callback function when the Android system fires an accessibility event. Such events include most UI state transitions by the user (agent) interactions, such as button clicking, window changing, etc, which fulfills the needs of the proposed benchmark.

The overview of MobileAgentBench is shown in Fig. 2. The benchmarking apps run on real devices from a device farm, which can either be a physical device or an emulator. The device talks to the host machine of the benchmark and the agent via ADB. The benchmark and the agent use ADB to retrieve app state information, such as screenshots, view hierarchy, and send control signals. The benchmark invokes the agent with the current benchmarking task prompt and collects runtime information from the agent, such as the LLM input and output. Whenever the event listener app receives an app event, it forwards the event to the benchmark server via a socket, so the benchmark can assess the task success status using both VH and the actions. At the top of Fig. 2, we show a sample task workflow, "Create a new task *Laundry*" with the Calendar app. The agent needs to perform 4 actions to complete the task: clicking the add button, clicking the task button, typing the task name "Laundry", and clicking the save button. The benchmark checks the content of the task name input box view and listens to the save button clicking event to determine if the task is finished successfully.

## 3.2 TASK DESCRIPTION

In our initial version of the benchmark, we implement 100 tasks over 10 daily apps. The 10 apps are from SimpleMobileTools [4], an open-source project of Android apps. These apps have simple and straightforward user interfaces, without any advertisements or unnecessary permissions, and thus are great for benchmarking use cases. The full list of app names is shown in Fig. 3b.

We carefully design the benchmarking tasks, so that they can simulate a normal user's daily activity and have multi-level difficulties. The distribution of task difficulty levels is shown in Fig. 3. Difficulties are defined by the minimum steps to finish a task, which is cross-verified by 3 human experts independently. A task would be classified as an *easy* task if it can be done within 2 steps, *medium* if greater or equal to 3 while less than 6, and otherwise, *hard*.

## 3.3 USAGE

The benchmark APIs are designed to be user-friendly and as less invasive as possible. For a standard agent, it takes less than 10 lines of additional code to integrate. List. 1 shows the pseudo-code of the benchmark usage. First, we need to import the benchmark Python library and initialize the benchmark orchestrator. Next, the main agent entrance function should be defined. It takes and

---

[4]  https://simplemobiletools.com, GPL-3.0

only takes one parameter, the task prompt. The agent iteratively performs actions to finish the task. Before and after the agent performs one action, the orchestrator functions are called, so information such as the time spent and LLM output can be collected. The program starts with the orchestrator run function. It calls the agent entrance function with each task prompt and automatically switches to the next task after the task finishes. The task success status is judged on the fly.

```python
from mobile_agent_benchmark.task_orchestrator import TaskOrchestrator
orchestrator = TaskOrchestrator() # the MobileAgentBench orchestrator
# the agent entrance function
def agent_run(task_prompt):
    while not done:
        orchestrator.before_one_action()
        # the agent invokes a LLM to think about the next action
        action = llm_think(task_prompt, screenshot)
        agent_perform(action)
        orchestrator.after_one_action(action)
orchestrator.run(agent_run)
```

Listing 1: Pseudo code of integrating MobileAgentBench into an existing Mobile LLM Agent.

## 4 EXPERIMENTS AND AGENT EVALUATIONS

### 4.1 METRICS

We define 6 metrics to comprehensively benchmarking mobile agents:

- Success Rate (SR): $SR = N_{success}/M_{tasks}$, where $N_{success}$ is the number of successful tasks, judged by the benchmark system. $M_{tasks}$ is the number of total benchmarking tasks. This metric reflects the agent's ability to correctly finish a task end-to-end.

- Step-wise Efficiency (SE). $SE = S_{actual}/S_{min}$, where $S_{actual}$ is the number of actual steps the agent takes to successfully finish a task, and $S_{min}$ is the minimum number of steps. This metric tells us if the agent performs unnecessary or redundant actions and reflects the efficiency of the agent. Failure tasks are not taken into account.

- Latency. The average time spent in seconds before and after one action. This metric tells us how long a user needs to wait between two actions.

- Tokens. The number of LLM input and output tokens. For simplicity, we use the GPT-4V (`gpt-4-vision-preview`) standard (OpenAI) to calculate the number of tokens for all models, which gives us a rough estimation of the LLM cost. For text, 1 token is 4 characters. For an image, it's divided into multiple $512 \times 512$ tiles, and each tile is 170 tokens. 85 base tokens are applied to each image as well.

- False Finish Rate (FFR). $FFR = N_{early}/M_{failure}$, where $N_{early}$ is the number of early stopped tasks and $M_{failure}$ is the total number of failure tasks. This metric represents how likely the agent falsely thinks it has finished the task and stopped early.

- Over Execution Rate (OER). $OER = N_{late}/M_{success}$, where $N_{late}$ is the number of late stopped tasks and $M_{success}$ is the total number of successful tasks. Symmetricly to FFR, this metric reveals how likely the agent falsely thinks the task is not finished successfully.

### 4.2 ENVIRONMENT SETUP

Five popular mobile LLM agents, AndroidArena (Xing et al., 2024), AutoDroid (Wen et al., 2023), AppAgent (Yang et al., 2023b), CogAgent (Hong et al., 2023), and MobileAgent (Wang et al., 2024) are evaluated with the proposed benchmark. We choose the Google Pixel 3a emulator and Android 14 operating system to run the benchmarking apps. Besides, the Android 9 operating system is used for AutoDroid as some of the dependency libraries do not support the newer Android systems.

As there are no local neural networks used in AndroidArena, AutoDroid, and AppAgents, these agents are executed on an Apple Macbook Pro with the M1 Max chip. CogAgent and MobileAgent require local model referencing, so they are executed on a workstation equipped with a single Nvidia RTX 4090 GPU, with 24 GB GPU memory.

Table 2: Agent performance results with multiple metrics

| Agent | Models | SR ↑ | SE ↓ | Latency ↓ | Tokens ↓ | FFR ↓ | OER ↓ |
|---|---|---|---|---|---|---|---|
| AndroidArena (Xing et al., 2024) | GPT4-V (Achiam et al., 2023) | 0.22 | **1.13** | 18.61 | 750.47 | **0.09** | 0.33 |
| AutoDroid (Wen et al., 2023) | GPT3.5 Instructor (Su et al., 2022) | 0.27 | 3.10 | **4.85** | 963.48 | 0.93 | **0.01** |
| AppAgent (Yang et al., 2023b) | GPT4-V (Achiam et al., 2023) | **0.40** | 1.29 | 26.09 | 1505.09 | 0.17 | 0.40 |
| CogAgent (Hong et al., 2023) | CogVLM (Wang et al., 2023a) | 0.08 | 2.42 | 6.76 | **579.84** | 1.00 | 0.04 |
| MobileAgent (Wang et al., 2024) | GPT4-V (Achiam et al., 2023) GroundingDINO (Liu et al., 2023) ViT-B/32 (Radford et al., 2021) DamoOCR (Wang et al., 2022) | 0.26 | 1.33 | 15.91 | 1236.88 | 0.19 | 0.31 |

The self-exploration feature is turned on for AppAgent. When performing a task, it can reference the previously summarized document. For CogAgent, we use 4-bit quantization to load the model due to GPU memory limitation. CogAgent is implemented in its vanilla flavor, *i.e.*, given the current screenshot, ask for the next action. No history information is provided.

### 4.3 RESULTS

The main experiment results are shown in Tab. 2. We observe that AppAgent has the highest success rate, benefiting from the self-exploration mechanism. CogAgent has the lowest success rate, most likely caused by the naive agent implementation, which limits the usage of history information. Although AutoDroid has a similar success rate to MobileAgent, the step-wise efficiency is significantly lower, possibly caused by the weaker reasoning capability of the GPT-3.5 model used by AutoDroid. Latency-wise, both AutoDroid and CogAgent have very low latency, indicating the high inferencing cost with the GPT-4V model. AppAgent needs to look up the app document, thus consuming more tokens than others. On the other hand, because of the naive agent implementation of CogAgent, it consumes the least number of tokens. AutoDroid and CogAgent have very high FFR, indicating they always stop early when the task is not finished yet. AppAgent, although having the highest task success rate, is not good at determining the task success, it cannot stop gracefully after finishing a task and has a high OER.

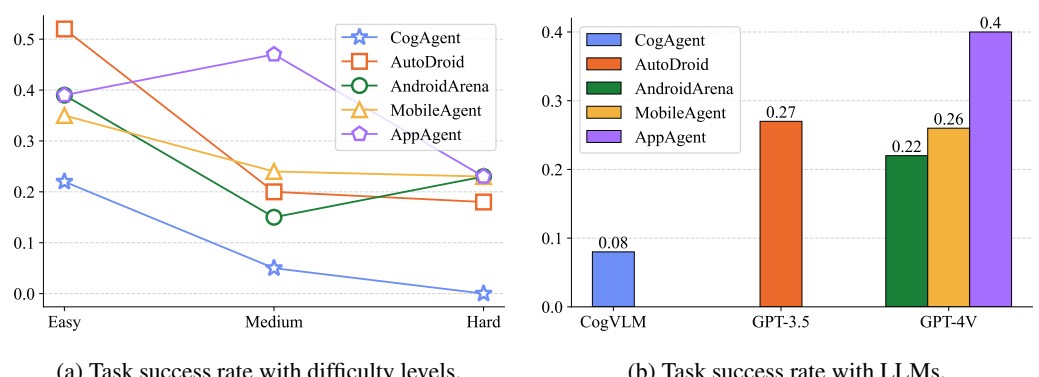

(a) Task success rate with difficulty levels.

(b) Task success rate with LLMs.

Figure 4: Task success rate.

The task success rates for each agent with difficulty levels are shown in Fig. 4a. From Fig. 4a, we can see most agents have higher success rates when handling easier tasks, which is as expected.

Interestingly, AppAgent has a higher success rate when performing medium-level tasks. This is because we set the maximum execution steps as twice the minimum steps, which would make the agents have very limited steps to correct their earlier steps for easy tasks. For example, an agent can only use 1 additional step to correct and finish the task for a 1-step easy task. However, for medium and hard tasks, there is significantly more space to correct the previous actions.

Fig. 4b shows the averaged task success rate over the backbone LLM models. It is interesting that AutoDroid, although using a text-based GPT-3.5 model, outperforms some other agents that use the more advanced GPT-4V model. This reveals that the textual view hierarchy contains the most important information for GUI navigation tasks. However, we believe that visual screenshots are helpful for other types of tasks, for example, if the task involves recognizing an image on the screen, or if the textual view hierarchy is not available.

## 5 LIMITATIONS AND FUTURE WORK

While the proposed MobileAgentBench is efficient, user-friendly, and addresses many issues of the existing benchmarks for mobile LLM agents, there are two main limitations that the authors would like to improve in the future. Firstly, although the use of Python code snippets as the configuration of task success conditions is easy for researchers in the AI/ML community, it is still difficult for people without a technical background. We will explore new methods to automatically build the task configuration without writing any code in the future work.

## 6 CONCLUSION

In this paper, we propose a new benchmark, MobileAgentBench, for mobile LLM agents on the Android platform. With the 100 built-in benchmarking tasks, researchers can test and evaluate existing and new agents automatically on real Android devices. Extending the benchmark to support customized tasks is also easy, as only basic Python coding skills are needed. Leveraging the Android Accessibility Services and only checking the final app state, MobileAgentBench can detect task completion status faithfully. We report the evaluation results of 5 popular agents across multiple metrics, and they can be used as strong baselines to advance future mobile LLM agent development.

## 7 REPRODUCIBILITY

Our source code is anonymously available at https://anonymous.4open.science/r/mobile-agent-bench-E727.

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

# A APPENDIX

## A.1 BENCHMARKING APPS

We choose 10 open-source daily apps from SimpleMobileTools[5] in our initial benchmark, a collection of simple Android apps without advertisements or unnecessary permissions. However, users can easily add their customized tasks with other apps. The reason that we choose open-source apps is that they are static and reproducible. Commercial apps often employ online A/B testing (**?**) mechanisms and may display UI elements and functionalities differently to different users, or change them over time.

The full list of the currently built-in apps is shown as follows:

- Calculator
- Calendar
- Contacts
- FileManager
- Gallery
- AppLauncher
- Messager
- MusicPlayer
- Notes
- Recorder

## A.2 CUSTOMIZATION

The 100 built-in tasks make the benchmark work out of the box. However, third-party researchers may want to implement their customized tasks. For example, benchmarking their agent performance on commercial apps. This can be done easily without modifying the benchmark library. First, implement the customized task class inheriting the Task class provided by the benchmark framework. The most important method to implement is `check_finish`, which defines how to judge if the given task is finished successfully. If necessary, the `setup` and `teardown` function can also be implemented to prepare and reset the environment to execute the current task. Then, add the task names to a JSON file and pass it to the benchmark's task orchestrator. When running, the task orchestrator will invoke each task (built-in or customized) one by one, automatically.

## A.3 EXPERIMENTS WITH ONLINE APPS

In this paper, we use open-source and offline apps because we want to create a fully controlled and reproducible environment for all agents. We picked apps such as Calendar and SMS that are commonly used by people for everyday tasks. Online apps, on the contrary, have dynamic content, which may make the agent's performance non-deterministic. Furthermore, even if the online app versions are the same (e.g. installing apps with the same APK), modern apps often have online A/B testing systems, which means the functionality or even UI appearance may not be the same for every user.

However, some researchers might be interested in benchmarking their agents with online apps. The proposed benchmark is designed to be easy to extend, so researchers can freely add their customized tasks with online apps. To demonstrate the extensibility of MobileAgentBench, we have implemented 6 tasks across 3 popular online apps: CNN News, Amazon Shopping, and Google Maps. Here is the full list of the currently implemented tasks.

- [Google Map] navigate from Bellevue, WA to Redmond, WA
- [Google Map] search Redmond, WA
- [Amazon Shopping] search nike and then show me shoes

---

[5] https://simplemobiletools.com

- [Amazon Shopping] show me the size guide for nike shoes
- [CNN News] show news under the climate section
- [CNN News] enable the CNN sound effect when opening the app

Tab. 3 shows experiment results on the subset of online apps. The metrics are the same as defined in Sec. 4.1.

Table 3: Agent performance results on Online Apps

| Agent | SR ↑ | SE ↓ | Latency ↓ | Tokens ↓ | FFR ↓ | OER ↓ |
|---|---|---|---|---|---|---|
| AndroidArena (Xing et al., 2024) | 0.33 | 2.40 | 13.95 | 2305.31 | **1.00** | 0.50 |
| AutoDroid (Wen et al., 2023) | 0.00 | N/A | **2.49** | **524.96** | N/A | **0.00** |
| AppAgent (Yang et al., 2023b) | **0.50** | 1.78 | 19.8 | 1518.62 | 0.33 | 0.33 |
| CogAgent (Hong et al., 2023) | 0.17 | 2.33 | 7.23 | 587.91 | **1.00** | **0.00** |
| MobileAgent (Wang et al., 2024) | **0.50** | **1.33** | 13.65 | 1225.67 | 0.33 | **0.00** |

## A.4 TASK SUCCESS CONDITION

List 2 shows a simple example of the `check_finish` function. To eliminate the difficulty of dealing with multiple possible action paths to success, we determine if the task is successful by only observing the final state. In this example, the task is to open the About page of the current app. We check if the Frequently Asked Questions text view (a child view in the About page) exists in the current UI page. If it exists, it means the About page is opened successfully and the task is completed. Otherwise, the task is not completed yet. It gives faithful results no matter how many ways there are to open the About page.

MobileAgentBench automatically invokes the `check_finish` function at the proper time: after one agent action, and when getting one app event. The `check_finish` function can be either stateless or stateful, depending on the task. The function has two parameters, the `view_client` and the `app_event`. `view_client` provides a set of useful functions to obtain the current page's view hierarchy. `app_event` provides the most recent user action sent from the Android Accessibility Service, such as a button-clicking event. It helps to check the task-finishing status when it's not directly reflected in the current UI view.

```python
def check_finish(self, view_client, app_event) -> bool:
    title_views = view_client.findViewWithText("Frequently asked
    questions")
    if title_views is not None:
        return True
    return False
```

Listing 2: A simple example of the `check_finish` function

## A.5 VIEW HIERARCHY AND ACCESSIBILITY EVENTS

MobileAgentBench utilizes AndroidViewClient, a popular Android automation framework, to obtain view hierarchy. AndroidViewClient has multiple backends, including UIAutomator, ViewServer, and CulebraTester2. In this paper, we use the UIAutomator as the backend to conduct experiments. However, it's worth exploring other backends that may have better performance. After getting the view hierarchy, the benchmark can access the properties of the views displayed on the screen. This provides a strong signal to check the current app status. For example, the benchmark can check if the query was correctly entered in a search bar.

To listen to the Android UI events, including button clicks, page changes, etc., we implement an Android app, AndroidEventListener. It uses the AccessibilityService and registers an event handler. Before launching the benchmark, the AndroidEventLinstener needs to be turned on on the system

settings, accessibility page. Then the process runs in the background and does not affect the foreground testing apps. Once an event is captured, it serializes the event object into JSON and sends it to the benchmark server using a socket. The benchmark consumes all events one by one after the agent performs an action.

## A.6    TASK DESCRIPTION

The full list of tasks is shown in Tab. 4. All tasks are carefully designed to reflect the functions of the corresponding app, with different difficulty levels. The difficulty levels are determined by the minimum steps to accomplish the task.

Table 4: Sample task prompts with their difficulty levels

| App Name | Difficulty Level | Task Prompt |
| --- | --- | --- |
| Calculator | Medium | Calculate the result of '3 + 5' |
| Calculator | Medium | Use Unit converter function to calculate how many kilometers 1mile is equal to |
| Calculator | Medium | Calculate the result of '3 ^ 3' |
| Calculator | Medium | Calculate the result of '12 ÷ 3' |
| Calculator | Medium | Calculate the result of '12 - 4' |
| Calculator | Hard | Calculate the result of '18+(24×3)-(9÷3)' |
| Calculator | Medium | Calculate the result of '12*4' |
| Calculator | Hard | Calculate the result of '19.7 - 81.3' |
| Calculator | Medium | Calculate the result of '12 × 5'. However, during the input process, the number '4' was mistakenly entered instead of '5'. Correct this by first enter 'C' to delete '4' and re-entering '5' and then perform the calculation |
| Calculator | Medium | Calculate the result of '$\sqrt{16}$ + 3' |
| Calendar | Medium | Create a new event 'laundry' and then search for it |
| Calendar | Easy | switch to daily view |
| Calendar | Medium | Show events in simple event list, delete the laundry and meeting events. |
| Calendar | Medium | Create a new event 'laundry' |
| Calendar | Medium | Create a new task, named 'laundry', with the description of 'wash all my clothes'. Mark it as all-day. |
| Calendar | Easy | show events of next month |
| Calendar | Easy | open about page |
| Calendar | Easy | search event 'laundry' |
| Calendar | Medium | change snooze time to 1 minute |
| Calendar | Medium | go to settings and make weeks start on Monday |
| Contacts | Easy | open About View |
| Contacts | Medium | Create a new contact, his First Name is Yuzai, and his Phone Number is 123456789 |

| Contacts | Medium | Delete contact Yuzai |
| Contacts | Easy | Set the contact Yuzai to Favorite |
| Contacts | Medium | Change phone filter, which means don't show phone storage in contacts view |
| Contacts | Medium | Set not show contact's Email in the contact profile screen |
| Contacts | Medium | Change the contact Yuzai's number to 987654321 and save it |
| Contacts | Medium | Remove the dialog button, and then return to the main view |
| Contacts | Easy | Search contact Yuzai |
| Contacts | Medium | Sort contacts by Data created time, Descending |
| File Manager | Easy | open the folder 'Downloads' and check the properties of the file 'testfile.txt' |
| File Manager | Easy | check the storage page |
| File Manager | Medium | Create a new file named 'testfile.txt' in the 'Downloads' folder |
| File Manager | Medium | Delete file named 'testfile.txt' in the 'Downloads' folder |
| File Manager | Medium | Delete the txt file in Download folder |
| File Manager | Hard | Delete all videos in Download folder |
| File Manager | Medium | Hide the folder named 'hidden' and make sure File Manager Stop showing hidden media |
| File Manager | Medium | open the folder 'Downloads' and rename the file 'Testfile.txt' to 'testfile1.txt' |
| File Manager | Easy | Search a file named 'testfile.txt' |
| File Manager | Medium | In the main page, sort the folder by size in descending order |
| Gallery | Medium | filter media in the gallery and only show images and videos |
| Gallery | Medium | Go to the downloads folder, group the images by file type |
| Gallery | Medium | Change the view type to list view |
| Gallery | Medium | Go to Gallery settings and enable play videos automatically |
| Gallery | Medium | Enable remember the last video playback position in settings |
| Gallery | Medium | Go to Downloads Folder and set the first image as favorite |
| Gallery | Medium | Go to Downloads Folder and set the first image as Home screen wallpaper |
| Gallery | Medium | show hidden items in the gallery in settings |
| Gallery | Medium | sort the gallery by size ascendingly |
| Gallery | Medium | Change the date and time format to 24-hour format in gallery settings |
| App Launcher | Medium | Add Chrome and Camera to launcher |
| App Launcher | Medium | Check who is the contributor of the app |
| App Launcher | Easy | Hide app name in Launcher |
| App Launcher | Medium | Open About page and go Frequently Asked Questions |
| App Launcher | Medium | Remove Chrome from Launcher |
| App Launcher | Medium | Rename Chrome in Launcher to MyChrome |

| App Launcher | Easy | Search for Chrome in Launcher |
|---|---|---|
| App Launcher | Medium | Change Setting Close this app at launching a different one to false |
| App Launcher | Medium | Sort apps by custom |
| App Launcher | Medium | Sort apps by title descending |
| SMS Messenger | Medium | Add a number '123456789' to block list |
| SMS Messenger | Medium | Change the Font size to 'Large' in the settings interface |
| SMS Messenger | Medium | open the conversation with contact number '123456789', and check for a random message's properties |
| SMS Messenger | Hard | start a conversation with number '123456789', send a message 'i luv u', and check for message properties |
| SMS Messenger | Hard | start a conversation with number '123456789', and send a message 'i luv u', back to the main page and search for the contact '123456789' |
| SMS Messenger | Medium | make the conversation with number '123456789' archived |
| SMS Messenger | Easy | search for the contact '123456789' at top search bar |
| SMS Messenger | Easy | search message 'i luv u' at the top search bar |
| SMS Messenger | Easy | show me the archived conversations |
| SMS Messenger | Hard | start a conversation with number '123456789', and send a message 'i luv u' |
| Music Player | Medium | sort the album by 'year' |
| Music Player | Medium | config equalizer to Heavy Metal |
| Music Player | Medium | create a new playlist:test |
| Music Player | Hard | create a new playlist: test, and search for it |
| Music Player | Hard | create a new playlist: test, and sort all playlist by descending order |
| Music Player | Medium | open faq page |
| Music Player | Easy | open setting page |
| Music Player | Medium | sort the playlist by 'desc' |
| Music Player | Easy | rescan media files |
| Music Player | Easy | search playlist 'Test' |
| Notes | Medium | add a new note named 'TODO List' |
| Notes | Medium | delete the "to_do_list" and "meeting" note |
| Notes | Easy | Check the item 'eggs' for shopping_list |
| Notes | Medium | use the pin '2580' to open the locked note 'password_list' |
| Notes | Medium | add a new Checklist named 'TODO List' |
| Notes | Hard | create a checklist named 'Shopping list' and add an item named 'Milk' |
| Notes | Easy | open the note 'meeting' |
| Notes | Easy | open about page |
| Notes | Medium | rename the current note to 'finished_task' |
| Notes | Easy | search 'secret' in note 'Charles's secrets' |

| Voice Recorder | Medium | change bitrate to 32 kbps |
| Voice Recorder | Medium | delete the last recorded audio |
| Voice Recorder | Hard | delete all recorded audio |
| Voice Recorder | Medium | go to settings and empty the recycle bin |
| Voice Recorder | Medium | use mp3 as the format for new recordings |
| Voice Recorder | Easy | go to recycle bin page |
| Voice Recorder | Medium | change settings, so that the deleted items will not go to recycle bin |
| Voice Recorder | Medium | rename the first audio to 'voice.m4a' |
| Voice Recorder | Hard | rename all audio to voice1.m4a, voice2.m4a, and so on |
| Voice Recorder | Medium | change app theme to dark red |

