# OpenReview forum: "MobileAgentBench: An Efficient and User-Friendly Benchmark for Mobile LLM Agents"
_ICLR.cc/2025/Conference — ICLR 2025 Conference Withdrawn Submission_

### Official Review · Reviewer_NJf2 · 2024-11-03

**Soundness:** 3
**Presentation:** 3
**Contribution:** 2
**Rating:** 6
**Confidence:** 4

**Summary:**

The paper presents MobileAgentBench, a benchmark for mobile LLM agents within the Android system, with a fully autonomous and reliable evaluation process. MobileAgentBench features itself for it can be run on real devices and needs no significant code changes to integrate agents into the framework.

The authors evaluate the performance of current SOTA mobile LLM agents on the new benchmark and find the success rates are relatively low, leaving space for further exploration.

**Strengths:**

1. Compared to the existing mobile LLM agent benchmarks, MobileAgentBench can be run on real Android devices to test mobile digital agents, making the evaluation process dynamic and realistic.
2. The benchmark can be extended and integrated with more task instances easily, with only several lines of Python codes needed. It is possible that the benchmark will attract more people in the community to contribute to scale and enrich it.
3. The evaluation process only checks the final state in the app system to detect if it is successfully completed, while allows the variance of trajectories/steps. It is effective and efficient way of evaluate digital tasks.

**Weaknesses:**

1. MobileAgentBench consists of 100 tasks totally, spanning across 10 simple use mobile apps with simple and straightforward user interfaces, which may damage the diversity and of the benchmark. The apps might need more careful selection and filtering, and the tasks could be more realistic as it would run into various situations in the wild environment.
2. The analysis part within the article is found to be inadequate and lacking in robustness. To enhance its credibility and depth, a more thorough comparative evaluation of agent performance is required. Furthermore, incorporating an examination of task types that span multiple applications could provide valuable insights.

**Questions:**

1. As the suceess rates of agents on MobileAgentBench are low, a detailed and thorough error analysis is needed to illustrate why they perform not well when doing the tasks and what is the bottleneck of improving on the benchmark. I recommend presenting some failure examples to better illustrate it.
2. How do you design and formulate the action space of MobileAgentBench? Could you explain more baout it?
3. In the part when you discuss about the digital assistant and human-computer interaction, I think one missing citation would be OSWorld[1], which is a benchmark of real-world computer tasks in a unified and interactive computer environment for multimodal agents. The task simulation and evaluation pipeline are quite related to your work.

[1] Osworld: Benchmarking multimodal agents for open-ended tasks in real computer environments. CoRR, abs/2404.07972, 2024. doi: 10.48550/ARXIV.2404.07972. URL https://doi.org/10.48550/arXiv.2404.07972.

---

### Official Review · Reviewer_YVMa · 2024-11-05

**Soundness:** 3
**Presentation:** 2
**Contribution:** 2
**Rating:** 5
**Confidence:** 4

**Summary:**

This paper introduces MobileAgentBench, a benchmark specifically designed to evaluate mobile LLM agents. The present work is motivated by the growing popularity of LLM agents in mobile setting, and the difficulty to develop a common platform to evaluate these diverse agents. The authors propose to alleviate the burden of manual setting by allowing LLM agents directly interact with GUIs. They 100 tasks across 10 open-source applications, categorizing them by difficulty to facilitate thorough evaluation. The experiment results highlight the strengths and weaknesses of existing agents, including AppAgent and MobileAgent.

**Strengths:**

1.The proposed benchmark addresses an important gap in evaluating mobile LLM agents, which is essential for the advancement of this field.

2.The authors implement 100 benchmarking tasks categorized by different levels of difficulty. It could provide a more nuanced evaluation of LLM agents.

3.The paper provides a systematic comparison of multiple agents, establishing a foundation for understanding their performance capabilities and limitations.

**Weaknesses:**

1.The scale and scope of designed tasks are generally smaller than other existing benchmarks. For example, SPA-bench [1] contains 340 tasks with both single APP and cross APP settings. Comprehensive comparison and discussion are needed to justify the unique advantage of the present work.

2.The paper lacks a detailed explanation of the underlying protocol of implementing MobileAgentBench, which could hinder reproducibility and applicability.

3.The evaluation metric of computation cost is limited. The authors only evaluate the token cost of using cloud-based LLMs, which do not consider the constrained computation resource in mobile devices.

4.The experiments do not include agents run on locally deployed LLMs like Phi-3 [2]. It misses an important setting for mobile agent, which has unique advantage for privacy preservation, low latency, etc.

[1] Chen, Jingxuan, et al. "SPA-Bench: A Comprehensive Benchmark for SmartPhone Agent Evaluation." NeurIPS 2024 Workshop on Open-World Agents. 2024.

[2] Abdin, Marah, et al. "Phi-3 technical report: A highly capable language model locally on your phone." arXiv preprint arXiv:2404.14219 (2024).

**Questions:**

Please refer to the Weaknesses above.

---

### Official Review · Reviewer_VQ1A · 2024-11-07

**Soundness:** 3
**Presentation:** 2
**Contribution:** 3
**Rating:** 3
**Confidence:** 3

**Summary:**

The paper introduces MobileAgentBench, an efficient and user-friendly benchmarking framework designed for evaluating large language model (LLM)-based mobile agents within the Android environment. Addressing limitations in existing benchmarks, MobileAgentBench allows for the autonomous testing of agents on real devices, with minimal code requirements for integration. The benchmark supports 100 predefined tasks across ten open-source applications, covering various difficulty levels. It evaluates agent performance based on metrics such as success rate, efficiency, and latency, and incorporates Android Accessibility Services to capture real-time app events. This design facilitates customizable and accurate testing, providing a robust platform for developing and evaluating intelligent mobile agents. The benchmark is interesting for mobile use cases. The writing needs to be further improved before acceptance consideration.

**Strengths:**

1. It is a rather new benchmark designed for mobile use cases.

2. The benchmark designs multiple tasks ranging from different difficulty levels.

3. It is easy-to-use, and can be integrated within few lines

**Weaknesses:**

1. Writing can be further improved

2. Large blank space should be fixed

**Questions:**

1. Writing of the paper can be further improved. For example, we do not need such detailed illustrations on related work as within this paper. Save the space for more benchmark analysis is preferable.

2. Your limitations and further work does not sufficiently comprise a single section. Merging it into conclusion is better.

3. Unnecessary blank space should be fixed such as from line 69 to line 74 and line 246 to line 249, etc.

---

### Official Review · Reviewer_boVc · 2024-11-09

**Soundness:** 2
**Presentation:** 2
**Contribution:** 2
**Rating:** 5
**Confidence:** 4

**Summary:**

This paper introduces MobileAgentBench, a new benchmark for evaluating Large Language Model (LLM)-based mobile agents on the Android platform. The authors argue that existing benchmarks suffer from limitations in scalability, robustness, flexibility, and realism.  MobileAgentBench aims to address these issues by providing 100 built-in tasks across 10 open-source Android apps, facilitating automated evaluation on real devices, and incorporating a flexible task success judgment mechanism based on final UI state and app event signals.  The benchmark also allows for easy customization and integration with existing agents, requiring minimal code modifications.  The authors evaluate five popular mobile LLM agents (AndroidArena, AutoDroid, AppAgent, CogAgent, and MobileAgent) using their benchmark and provide baseline performance data.

**Strengths:**

The authors convincingly identified important dimensions in current mobile agent benchmarks, particularly regarding scalability, robustness to diverse action sequences, realistic device-based testing, and ease of integration. Here are some strengths.

Automated evaluation: The framework automates the evaluation process, reducing manual effort and increasing reproducibility.
Accessibility and ease of integration: The benchmark is designed to be easily integrated with existing mobile agent frameworks, requiring minimal code changes.
Open-source and reproducible:  The authors commit to making the benchmark open-source, promoting transparency and further development by the community.
Baseline data provided: The evaluation of five existing agents offers valuable baseline data for future research.

**Weaknesses:**

I think main motivation for building a new benchmark should ultimately be about "can we evaluate better" or "can we evaluate more complex tasks". While suggested benchmark seem more "user-friendly" than the referenced ones, I'm not quite convinced if MobileAgentBench is moving us forward.

Limited app diversity results limited agent behaviors. The benchmark currently relies on 10 open-source apps from a single developer (SimpleMobileTools).  While understandable for initial development, this limits the diversity of UI elements, interaction patterns, and complexities that agents face. The provided tasks, while covering basic functionalities, might not adequately capture the complexity of real-world mobile interactions.  I expect a new benchmark that encompasses the referenced ones to involve more intricate, multi-step tasks involving data input, navigation across multiple apps, and handling errors are needed.

Limited metric depth:  While the proposed metrics (SR, SE, Latency, Tokens, FFR, OER) are relevant, they could be expanded to capture aspects like agent robustness to unexpected UI changes, error recovery, and efficiency in terms of actions taken.

Limited explanation of the agent event listener app: The functionality and implementation details of the Android Accessibility Service-based event listener app are not thoroughly explained. A more detailed description is crucial for understanding the robustness and reliability of the event capture mechanism.

**Questions:**

How does MobileAgentBench handle tasks that require interactions with system-level UI elements (e.g., notifications, permission requests)?

What is the specific implementation of the "hit test" used by the benchmark to determine successful button clicks?

How does the framework handle cases where the agent crashes or the app under test becomes unresponsive?

How does the choice of UIAutomator as the backend for AndroidViewClient impact performance and reliability?  Have other backends been considered?

---

### Note · Authors · 2024-11-25

I have read and agree with the venue's withdrawal policy on behalf of myself and my co-authors.